Exotic and native plants play equally important roles in supporting and structuring plant-hummingbird networks within urban green spaces

Sánchez Sánchez Monserrat 1
http://orcid.org/0000-0001-6037-0327 Lara Carlos 2 carlos.lara.rodriguez@gmail.com
1 Universidad Autónoma de Tlaxcala, Maestría en Biotecnología y Manejo de Recursos Naturales , San Felipe Ixtacuixtla, Tlaxcala , Mexico
2 Centro de Investigación en Ciencias Biológicas, Universidad Autónoma de Tlaxcala , Ixtacuixtla, Tlaxcala , Mexico
Measey John
Electronic publication date: 2024 Feb 21
Publication date: 2024
Volume: 12
Electronic Location ID: e16996
Received 2023 Oct 25; Accepted 2024 Feb 2
Copyright: © 2024 Sánchez Sánchez and Lara
Copyright year: 2024
Copyright holder: Sánchez Sánchez and Lara
License: This is an open access article distributed under the terms of the Creative Commons Attribution License, which permits unrestricted use, distribution, reproduction and adaptation in any medium and for any purpose provided that it is properly attributed. For attribution, the original author(s), title, publication source (PeerJ) and either DOI or URL of the article must be cited.
License URL: https://creativecommons.org/licenses/by/4.0/

Keywords: Exotic plants, Native plant-pollinator interactions, Hummingbirds, Network structure, Ecosystem resilience

Funding: Consejo Nacional de Ciencia y Tecnología (CONACyT) through the masters’ scholarship 1077502 This work was supported by the Consejo Nacional de Ciencia y Tecnología (CONACyT) through the masters’ scholarship (Number: 1077502). The funders had no role in study design, data collection and analysis, decision to publish, or preparation of the manuscript.

==============================
Background

Urban gardens, despite their transformed nature, serve as invaluable microcosms for a quantitative examination of floral resource provision to urban pollinators, considering the plant’s origin. Thus, knowledge has increased, emphasizing the importance of these green areas for hosting and conserving pollinator communities. However, there is a significant knowledge gap concerning the changing availability of these native and exotic floral resources over time and their impact on structuring interaction networks with specific pollinators.

Methods

Over a year-long period, monthly surveys were conducted to record both native and exotic plant species visited by hummingbirds in an urban garden at Tlaxcala, Mexico. Flower visits were recorded, and the total flowers on each plant visited were tallied. Additionally, all observed hummingbirds were recorded during the transect walks, regardless of plant visits, to determine hummingbird abundance. The interactions were summarized using matrices, and network descriptors like connectance, specializacion, nestedness, and modularity were computed. Plant and hummingbird species in the core and periphery of the network were also identified. Lastly, simulations were performed to assess the network’s resilience to the extinction of highly connected native and exotic plant species, including those previously situated in the network’s core.

Results

We recorded 4,674 interactions between 28 plant species, and eight hummingbird species. The majority of plants showed an ornithophilic syndrome, with 20 species considered exotic. Despite asynchronous flowering, there was overlap observed across different plant species throughout the year. Exotic plants like Jacaranda mimosifolia and Nicotiana glauca produced more flowers annually than native species. The abundance of hummingbirds varied throughout the study, with Saucerottia berillyna being the most abundant species. The plant-hummingbird network displayed high connectance, indicating generalization in their interaction. Significant nestedness was observed, mainly influenced by exotic plant species. The core of the network was enriched with exotic plants, while Basilinna leucotis and Cynanthus latirostris played central roles among hummingbirds. Network resilience to species extinction remained generally high.

Conclusions

Our findings provide valuable insights into the dynamics and structure of plant-hummingbird interactions in urban gardens, emphasizing the influence of exotic plant species and the network’s resilience to perturbations. Understanding and managing the impact of exotic plants on such networks is crucial for the conservation and sustainable functioning of urban ecosystems.

Introduction

The proliferation of exotic plant species in native ecological communities is a pressing global concern (Sax & Gaines, 2008). Anthropogenic activities such as changes in land use, habitat fragmentation, and intentional or unintentional species introductions often facilitate the successful integration of exotic plant species into these communities, disrupting their evolutionary and ecological balance (Traveset et al., 2013). Exotic species, being adept invaders, tend to outcompete native species, leading to cascading effects throughout the existing communities and forming novel communities that are often more resilient to restoration efforts (Lodge & Sgrader-Frechette, 2003).

Establishing exotic plant species faces a significant barrier: acquiring novel native floral visitors (Aizen, Morales & Morales, 2008; Bartomeus, Vilà & Santamaría, 2008; Díaz Infante, Lara & Arizmendi Mdel, 2020). These are a critical resource that both native and exotic plants compete for (Padrón et al., 2009). Research indicates that moderate changes in land use can enhance the richness and abundance of visitors by increasing habitat heterogeneity and resource availability, often due to the incorporation of exotic plant species (Tscharntke et al., 2008; Tews et al., 2004). Human-created habitats provide new opportunities for resource exploitation by floral visitors, showcasing their adaptability and flexibility in utilizing altered landscapes (Klein, Steffan–Dewenter & Tscharntke, 2003). However, a higher diversity of visitors in modified habitats may mask the negative effects of exotic species on the overall interacting community due to an abundance of exploitable plant species (Winfree, Bartomeus & Cariveau, 2011).

One notable impact of introducing exotic plant species is the altered attraction of floral visitors to native plants. Invasive exotic plants often possess traits that make them more attractive to visitors, including vivid colors, large corolla sizes, and abundant nectar rewards. This can lead to behavioral changes in native visitors, potentially reducing their visits to native flora (Lopezaraiza-Mikel et al., 2007; Gibson, Richardson & Pau, 2012; LeVan et al., 2014). However, these effects can also be neutral or even positive. The joint attraction of native and exotic plants can increase floral visitor arrival, enhancing pollination services for both types of plant species (Jakobsson & Padrón, 2014). The effects of exotic plants on native ones also depend on various factors such as floral abundance, visitor abundance, or community species richness (Morales & Traveset, 2009).

The impacts of exotic plant species on plant-floral visitor network structures have been subject to varying findings. Some studies have shown that exotic plant species can affect network specialization, modularity, nestedness, and robustness, while others have found little or no effect on network structure (Vilà et al., 2011; Kaiser-Bunbury et al., 2011; Albrecht et al., 2014). This variability may stem from differences in the amount of exotic floral resources available, known as invasion intensity, among study systems (Kaiser-Bunbury et al., 2011; Tylianakis & Morris, 2017). To comprehensively understand the consequences of exotic plant species on interactions between native plants and visitors, it is crucial to assess how these effects vary at different levels of exotic species richness and floral abundance (Olesen, Eskildsen & Venkatasamy, 2002; Maruyama et al., 2016).

Exotic species can induce the rewiring of potential links within an interaction network. Generalist native species may transfer their links to highly connected exotic species as the invasion progresses, altering the network structure and topology (Aizen, Morales & Morales, 2008; Kaiser-Bunbury et al., 2011; de M Santos et al., 2012). Exotic species are typically generalists, causing nesting, stability, and increased connectivity of network modules. For example, generalist plant invaders can either merge modules, creating larger ones with more species, or fragment the network into smaller modules by monopolizing native partners (Olesen et al., 2007; Padrón et al., 2009). They may establish new modules by usurping species or attracting additional pollinators. While the overall module number might stay the same, the boundaries may blur, improving connectivity and cohesion within interaction networks (Albrecht et al., 2014). However, this disruption in the balance of plant-floral visitor networks due to exotic plant species may impede the restoration of native species in a given site (Valdovinos, 2019; Traveset et al., 2013). Consequently, effective management of ecosystems necessitates considering the various interactions and relationships between species.

Pollinators are crucial for maintaining urban and peri-urban flora, contributing to fruit and seed production (Baldock et al., 2015; Hall et al., 2017). In urban areas, insects and birds, particularly hummingbirds (Trochilidae), are the primary pollinators, interacting with plants in diverse habitats, including urban green areas (Toledo & Moreira, 2008; Rodrigues & Araujo, 2011). Hummingbirds, often generalists in highly modified urban settings, demonstrate adaptability in resource utilization, including feeding on a variety of flower sizes and colors (Maruyama et al., 2013; Waser, CaraDonna & Price, 2018). In urban green areas, a mix of native and exotic plant species is common (Abella & Chiquoine, 2019), with some non-native plants attracting more pollinators, including insects (Morales & Traveset, 2009; Matteson & Langellotto, 2011). This suggests that non-native plant species may significantly influence network structure features in plant-hummingbird interactions.

Studies have begun exploring how the inclusion of non-native floral resources affects plant-hummingbird interaction networks. Research comparing urban and conserved natural areas revealed that generalist hummingbirds persist in urban landscapes, adapting to the presence of both exotic and native plants and artificial feeders (Maruyama et al., 2019). Another study demonstrated that both exotic and native plants, along with feeders, play essential roles in the structure of plant-hummingbird interaction networks, connecting natural and human-modified habitats (Ramírez-Burbano et al., 2021). Likewise, a study in a tropical urban environment revealed that highly connected and abundant native plant species played a crucial role in network stability, overshadowing the impact of exotic species (Vitorino, da Frota & Maruyama, 2021). However, caution is warranted, as the network studied had relatively few exotic species. These findings emphasize hummingbirds’ adaptability in integrating non-native resources into their interaction networks with native plants and highlight the need to further explore the consequences of this integration.

Recently, Parra‐Tabla & Arceo‐Gómez’s (2021) review has identified significant effects of exotic species on these networks, emphasizing alterations in interaction patterns, frequency, and individual species roles. However, despite acknowledging these effects, there remains a notable gap in comprehending the nuanced mechanisms underlying these alterations. Specifically, there is a lack of detailed insights into how these modifications are related to the temporal dynamics of native and exotic floral resources supplying specific pollinator assemblies. For example, while native floral cover fluctuates throughout the year, exotic plants can overlap and/or complement the flowering time of native plants through gardening practices, as commonly occurs in urban green areas (Erickson, Patch & Grozinger, 2021; Staab, Pereira-Peixoto & Klein, 2020). This phenomenon emphasizes the pivotal role of seasonal dynamics in shaping pollination networks and influencing species persistence (Zaninotto, Thebault & Dajoz, 2023). This deeper understanding is crucial not only for theoretical advancements in community ecology but also for developing more effective conservation and management strategies in the face of increasing plant invasions by exotic species.

Here, we monitored hummingbird-plant interactions over the course of a year in an urban green area in Tlaxcala, Mexico, quantifying the intensity of native and exotic flower use by hummingbirds as well as the abundances of the interacting species. Specifically, we aimed to determine which plant group, native or exotic flora, constituted the main nectar resource available for hummingbirds at the study site. Likewise, from a topological perspective (utilizing various metrics and simulating species extinction to highlight their role as network structurers), we examined the contributions of both native and exotic plants in structuring the interaction network with hummingbirds. Within the urban green area, the prevalence of exotic floral resources is expected to entice hummingbirds, potentially leading to a heightened reliance on these exotic plants for nectar. This flower use might significantly shape the interaction network, highlighting the pivotal roles of both native and exotic plants. As we investigate the core species and their removal, we anticipate insights into how disruptions to key exotic plants might reverberate through the network, potentially altering its stability and revealing dependencies of certain hummingbird species. These predictions aim to unravel the intricate balance between native and exotic plants and their impact on the resilience and structure of the hummingbird-plant interaction network within the urban setting.

Materials and Methods

Study area

Interactions between hummingbirds and their plants were monitored from May 2018 to April 2019 at the Tizatlán Botanical Garden (hereafter TBG) in Tlaxcala, México (19° 19′ 44–19° 19′ 57 N, and 98° 13′ 13–98° 12′ 49 W, at an elevation of 2,250 m above sea level). This urban green area was established in 1989 as an alternative to the existing environmental deterioration in the state, with the goal of conserving the floral heritage of the region. The area hosts collections of live plants, each labeled, registered, and documented for educational, cultural, and scientific research purposes, facilitating the rapid identification of plants used in this study (Acosta, 1990).

The TBG covers an area of eight hectares, divided into six sections: woodland, xerophilous scrub, tropical nursery vegetation and greenhouses, fruit and medicinal plants, and ornamental plants. Since its establishment, the integration of exotic plants has been common throughout the area covered by the urban garden. Additionally, a network of trails allows visitors to explore all these areas, aiding in the establishment of the transects described below (Acosta, 1991).

The climate in this zone is temperate sub-humid with summer rains. The average annual temperature ranges from 12 to 24 °C, and the average annual precipitation fluctuates between 700 and 1,000 mm (INAFED, 2010). Topographically, the site is nearly flat and is bordered to the northwest by the Zahuapan River and the Río de los Negros, with the rest of the land adjoining urban areas. The soil in the garden primarily consists of a fluvial sandy-loamy texture (Acosta, 1991). The dominant tree stratum of the location is characterized by the presence of alders (Alnus sp.) and willows (Salix sp.), as well as oaks (Quercus sp.), surrounded by small areas of xerophilous scrub.

Surverys of plant-hummingbird interactions

Before initiating the monitoring and with the aim of covering the main areas of this urban green space, three transects of 250 m in length were established (always utilizing visitor trails), with a separation of at least 100 m between them. Censuses of plant-hummingbird interactions were conducted monthly during clear and sunny conditions from May 2018 to April 2019 on each transect. Each transect was walked at a moderate pace, and all flowering plants visited by hummingbirds, whether directly on the imaginary line of the transect or within a parallel distance of 20 m, were recorded. Two trained observers walked the transects and conducted the censuses during hummingbird foraging activity from 08:00 to 12:00 h. All surveys were conducted by the same observers. To mitigate order effects, the starting points and directions of the transects were alternated throughout the study (RalpH, 1997). Binoculars (10 × 40 mm) and bird guides (Arizmendi & Berlanga, 2014) were used to identify the hummingbirds. Plant species observed were identified using their aforementioned labels and further confirmed through photographs. Likewise, plants were classified as either native or exotic according to openly available databases such as GRIN Taxonomy for Plants for North America (http://www.ars-grin.gov/) and World Flora Online (http://www.worldfloraonline.org/). During the surveys, the following data were recorded for each interaction between hummingbirds and plants: hummingbird species, plant species, and the number of flowers visited. After completing the surveys, the total number of flowers on each plant visited by hummingbirds was counted. All ornithophilous plant species available at each month in the study area were visited by hummingbirds through the surveys. Additionally, during each transect walk, every observed hummingbird, whether or not it visited a plant, was recorded to establish a measure of hummingbird abundance.

The completeness of the record of interactions between plants and hummingbirds was determined using the Chao2 estimator in the iNEXT package in R (Hsieh, Ma & Chao, 2013). Differences between hummingbird species and between plant types (native or exotic) in their abundances throughout the study were analyzed using generalized linear mixed models (GLMMs, package glmmTMB, Brooks et al., 2017) using a Poisson error distribution family and logarithmic link function in R 4.0.0 (R Core Team, 2020) to analyze our count data. In the models, hummingbird species, plant type (native or exotic), were used as factors, and the number of recorded hummingbirds and flowers were used as response variables. We then construct a similar model to examined the impact of plant type (predictor) on the number of hummingbird visits as response variable. Each of the above models had the sampling month as a random intercept effect.

To determine which flowers hummingbirds prefer, whether from native or alien plant species, we assessed the frequency of their visits to various flowers. To achieve this, we employed a metric known as the preference index (PI). This metric evaluates the relative number of visits a plant species receives in comparison to the relative number of flowers of that species, thereby retaining information about its relative abundance (Kells, Holland & Goulson, 2001). The metric utilizes a scoring system to rank all plant species, ranging from the most to the least preferred (for more details, see Pizante et al., 2023).

Interaction network analysis

We summarized interactions (i.e., hummingbird foraging visits to native and exotic plants) using a bipartite matrix, where each cell indicated the frequency of pairwise interactions between a plant and a hummingbird species. From the interaction matrix obtained, we constructed the interaction network using the Bipartite package in the “R” software (R Core Team, 2020). To assess and describe the topological properties of the plant-hummingbird network, we estimated the following metrics: connectance, network specialization, nestedness, modularity, identification of core/periphery species, and robustness. We chose these metrics because their values offer a comprehensive view of the organization and relationships within a community. (1) Connectance refers to the proportion of realized interactions (actual links or connections) relative to all possible interactions between different entities within the network, and their values range from 0 to 1, with 1 representing a fully connected network where every entity is interacting with every other entity, and 0 representing a network with no interactions. (2) Specialization (H) quantifies the degree to which species within the network are specialized in their interactions with other species, and ranges from 0 to 1, where 0 indicates a completely generalized (non-specialized) network, and one indicates a completely specialized network. (3) Nestedness, is a metric of the degree to which species in a network interact in a non-random way. In a nested network, the interactions of the species with fewer interactions are a subset of the interactions of the species with more interactions. We used the normalized nestedness metric, NODFc, based on the NODF measure: NODFc = NODFn/(C·log(S)), where NODFn = NODF/max(NODF), C is the connectance, S is the geometric mean of the number of species in each level of the network, NODF is the raw NODF value for the network and max (NODF) is the maximum nestedness of a network with the same number of species and links as the focal network, subject to the constraint that every species has at least one link (Song, Rohr & Saavedra, 2017). We used the maxnodf package to implement calculation of the NODFc metric (Hoeppke & Simmons, 2021). Given the significant nestedness of our network (refer to the Results section), we investigated the respective contributions of plants and hummingbirds to this nested pattern. We calculated the nestedness contribution (Nc), which reflects how the interactions of plant or hummingbird species either enhance or diminish the overall network nestedness. This was then compared to our randomized expectations, following the methodology outlined by Saavedra et al. (2011). Positive values of cni denote a stronger contribution of species i to the nested structure compared to what would be anticipated by chance. (4) Modularity (Q) is a measure that identify subgroups of nodes (e.g., species) that interact more frequently among themselves than with nodes in other subgroups. The QuaBiMo optimization algorithm (Dormann & Strauss, 2014), was employed for the quantitative matrix obtained. The modularity score ranges from −1 to 1. A higher positive modularity indicates a network with a clear modular structure, where nodes within a module have strong connections with each other but weaker connections with nodes outside their module. On the other hand, a score close to 0 or negative modularity suggests a network without a clear modular structure. To assess the statistical significance of connectance, specialization and modularity (except nestedness), we compared the observed values to 1,000 random values that were generated using a randomization algorithm called Patefield’s r2dtable algorithm, which conserves the total number of interactions per row and column in the matrix. We expressed these network indices as z-scores (observed-mean (null)/sd (null)), and evaluated the statistical significance using z-tests. (5) We also calculated the network core (Gc) using the following formula: Gc = (ki/kmean)/σk, where ki is the mean number of links for a given hummingbird or plant species, kmean is the mean number of links among all hummingbird or plant species in the network, and σk is the standard deviation of the mean number of links among all hummingbird or plant species. A Gc value greater than one indicates a core species, while values close to zero or less than zero can be considered as peripheral species (Dáttilo, Guimarães & Izzo, 2013).

Finally, based on the previous analyses, we selected the two main core species of native and exotic plants. One by one, we simulated the extinction of these species to record the changes in the robustness of the plant-hummingbird network structure obtained at our study site. Network robustness refers to the ability of an ecological network to maintain its structure, functionality, and stability in the face of perturbations, such as species extinctions. A robust network can withstand disruptions or species loss without undergoing significant changes in its overall structure or losing critical ecological functions. Robustness ranges from 0 to 1, where higher values indicate a more robust network (Burgos et al., 2007).

Results

Hummingbirds and their flowering plants

The plant-hummingbird data matrix comprised a total of 4,674 interactions among 28 plant species belonging to 20 families, and eight hummingbird species. Throughout the year, we detected 99.74% of the estimated possible interactions between plants and hummingbirds in the study area, according to the Chao2 estimator. Therefore, the number of observed interactions appeared to reach the asymptote with respect to our sampling effort. The list and abundances of plants and hummingbirds recorded throughout the study at the TBG are shown in Table 1 and Fig. S1 (see File S1). The majority of these species exhibit ornithophilous syndrome, and 20 of them are considered exotic. The flowering of these species is asynchronous, although there is overlap in their flowering throughout the year (Fig. 1A). The plant species that produce the highest number of flowers per individual are two exotic species: Jacaranda mimosifolia (mean ± SD = 273.83 ± 321.31) and Nicotiana glauca (263.58 ± 147.06); and one native species, the mistletoe Psittacanthus calyculatus (272.91 ± 337.09). Exotic plant species produce a greater number of flowers throughout the year compared to native plant species (z-value = 27.4, df = 1, P < 0.0001). However, their flowers receive a significantly lower number of visits by hummingbirds compared to the flowers of native plant species (z-value = 413, df = 1, P < 0.0002; Figs. 2A and 2B).

Table 1 List of native (N) and exotic (E) plant species and the number of flowers visited by hummingbirds.

Species	Family	Type	Hummingbird visits	
Abutilon pictum (Gillies ex Hook.) Walp.	Malvaceae	E	49	
Bouvardia ternifolia Schltdl.	Rubiaceae	N	469	
Citrus reticulata Blanco	Rutaceae	E	19	
Crocosmia crocosmiiflora (G. Nicholson) N.E.Br.	Iridaceae	E	59	
Cydonia oblonga Mill.	Rosaceae	E	60	
Cylindropuntia imbricata (Haw.) F.M. Knuth	Cactaceae	N	48	
Grevillea robusta A.Cunn. ex R.Br.	Proteaceae	E	285	
Impatiens balfourii Hook.f.	Balsaminaceae	E	66	
Jacaranda mimosifolia D. Don	Bignoniaceae	E	164	
Jasminum grandiflorum L.	Oleaceae	E	137	
Kniphofia linearifolia Baker	Asphodelaceae	E	136	
Malus domestica (Suckow) Borkh.	Rosaceae	E	38	
Melaleuca citrina (Curtis) Dum. Cours.	Myrtaceae	E	573	
Nerium oleander L.	Apocynaceae	E	84	
Nicotiana glauca Graham	Solanaceae	E	444	
Oenothera elata Kunth	Onagraceae	N	81	
Opuntia robusta Pfeiff.	Cactaceae	E	53	
Pittocaulon praecox (Cav.) H.Rob. & Brettell	Asteraceae	N	318	
Prunus domestica L.	Rosaceae	E	68	
Prunus persica (L.) Stokes	Rosaceae	E	172	
Prunus serotina Ehrh.	Rosaceae	N	151	
Psittacanthus calyculatus G. Don	Loranthaceae	N	562	
Punica granatum L.	Lythraceae	E	134	
Pyrus communis L.	Rosaceae	E	48	
Rubus ulmifolius Schott	Rosaceae	E	55	
Tecoma stans (L.) Kunth	Bignoniaceae	N	306	
Tulbaghia violacea Harv.	Amaryllidaceae	E	40	
Verbascum virgatum Stokes	Scrophulariaceae	E	55	
Note:

Plant species names and families recorded at the Tizatlán Botanical Garden, Mexico, following ‘World Flora Online’ (http://www.worldfloraonline.org).

Figure 1 Stacked area charts illustrate the monthly abundance of (A) native and exotic plants, as well as (B) hummingbirds, recorded over the course of 1 year.

The charts showcase the temporal variation in abundance for these ecological elements, providing insights into their seasonal dynamics and potential interactions at the Tizatlan Botanical Garden.

Figure 2 Variation of hummingbird visits to plants recorded at the study site.

Variation of hummingbird visits to plants recorded at the Tizatlan Botanical Garden. Violin plots showing the kernel probability density of the data at different values to illustrate (A) differences between native and exotic plant species in the number of hummingbird visits. The box plots within each violin plot denote the median and interquartile range along with outliers. (B) A White-eared hummingbird (Basilinna leucotis) visiting an exotic plant species (Melaleuca citrina). Photo credit: Carlos Lara

The above was also supported by the analyses of hummingbird preferences. Despite the great variation in the total number of open flowers among different plant species throughout the study, those species that were most numerous were not necessarily the ones that received the highest proportion of hummingbird visits. J. mimosifolia and N. glauca were the most numerous flowers throughout the year, but both exotic plant species did not score highly on the preference indices for hummingbirds when compared with species present at lower abundance, particularly native plant species. Interestingly, four out of the five plant species with the highest preference for hummingbirds were native species, with the most preferred being the perennial shrub Tecoma stans. Among the exotic species, Melaleuca citrina exhibited the highest preference. All preference values and ranks are shown in File S1.

The hummingbird assemblage was composed of both resident species (Saucerottia beryllina, Cynanthus latirostris, and Basilinna leucotis) and some species that are present only during certain months in the area (Leucolia violiceps, Selasphorus heloisa, Archilochus colubris, Calothorax lucifer, and Lampornis clemenciae). The abundances of these species fluctuated significantly throughout the year (z-value =180.15, df = 11, P < 0.001; Fig. 1B and File S1). The most abundant species during the study was Saucerottia beryllina (z-value = 395.75, df = 7, P < 0.001).

Plant-hummingbird interaction network

The plant-hummingbird interaction network at the TBG (Fig. 3) exhibited high connectance values (0.65, 0.35, z-score = −2.82, p = 0.005) and low levels of specialization H2’ (0.11, z-score = −46.17, p = 0.007), demonstrating ecological generalization in the interaction between plants and hummingbirds at this site. The Q value (0.147, z-score = 32.31, p < 0.001) indicated significant modularity, which was larger than expected by random chance. We obtained four modules: one formed by the hummingbirds Leucolia violiceps, Archilochus colubris, and Lampornis clemenciae; another by Cynanthus latirostris and Basilinna leucotis; one more by Selasphorus heloisa and Calothorax lucifer; and finally, one formed solely by Saucerottia berillyna. The modules were not segregated by the use of exotic or native plant species, but it is clear that all modules include exotic plant species. However, only one module stands out as the sole dominantly native plant module (Fig. 3).

Figure 3 Network of plants and their hummingbirds with identified modules indicated by colors.

Network of plants and their hummingbirds with identified modules indicated by colors. Ecological network obtained from hummingbird visits to native (N) and exotic (E) plant species through the study. The circle size denotes their importance as core species, and the circle colors reflect different species that belong to the same module. The thickness of each link (gray lines) indicates the frequency of each pairwise interaction (i.e., foraging). The circles for hummingbirds are labelled.

The structure of the plant-hummingbird network at the TBG was significantly nested (NODF = 76.59, z-score = 7.35, p < 0.001; wNODF = 60.47, z-score = 6.98, p = <0.001). Out of the 28 plant species visited by hummingbirds throughout the year, 13 showed a significant contribution to the nestedness of the interaction network (Fig. 4A). Among these, 84.6% (11 species) were exotic species, constituting 71.4% (20 out of 28 species) of the total species visited by hummingbirds at the study site. Regarding hummingbirds, four out of the eight species recorded during the study significantly contributed to the nestedness of the interaction network (Fig. 4B). Two of these are resident species in the area (Basilinna leucotis and Cynanthus latirostris), and two are migratory species that visit the area during specific months of the year, particularly in the fall-winter season (Leucolia violiceps and Archilochus colubris).

Figure 4 Contribution of plant and hummingbird species to the nested structure of the network at the TGB.

Contribution to nestedness by (A) native and exotic plant species, and (B) hummingbird species that integrate the plant-hummingbird network. Species are organized in decreasing order according to their contribution, and the names of the species with the greatest values are highlighted. Negative values indicate that the presence or interactions of that species decrease the overall nested structure of the network.

The identity of core and peripheral species in the interaction network showed a differential role of exotic and native plants. For instance, out of the 28 plant species in the network, seven are part of a dense and defined core. Among these, 57.14% (four species) are exotic plants, which include Tree Tobacco (Nicotiana glauca), Cajeput Tree (Melaleuca citrina), Jacaranda (Jacaranda mimosifolia), and Silky Oak (Grevillea robusta). Regarding native plants, the central role of mistletoe (Psittacanthus calyculatus), Locoweed (Pittocaulon praecox), and Firecrackerbush (Bouvardia ternifolia) stands out (Supplemental File). The remaining 21 plant species are peripheral in the interaction network and have limited importance for connectivity, both between and within the network modules. In the case of hummingbirds, two species are part of the core in the network: Basilinna leucotis and Cynanthus latirostris. These two species are considerably more connected in their interaction with the plants. On the contrary, the rest of the hummingbird species showed a lower number of connections within the network (Supplemental File).

The overall network robustness was high (R = 0.8727). From this overall network, the removal of the two core exotic plant species (Nicotiana glauca and Melaleuca citrina) did not affect the network’s robustness (R = 0.8529). A similar pattern was observed when removing the core native plant species (R = 0.8591). The fact that the robustness value remains unchanged when removing these species, whether native or exotic, indicates a certain strength or redundancy in the network. This enables it to withstand the loss of these species without undergoing a significant collapse in its structure or function.

Discussion

In the TBG, there are 20 exotic and eight native plant species interacting with hummingbirds. Asynchronous flowering occurs throughout the year, ensuring flower availability for these tiny birds. Many plants exhibit an ornithophilous syndrome. Specifically, resident hummingbird species, such as Basilinna leucotis and Cynanthus latirostris, interact more with plants at the study site. The interaction network exhibited a nested, modular structure, where hummingbird species interacted with both native and exotic plant species. Among the thirteen plant species contributing the most to this nested structure, eleven are exotic species, four of which are part of the network’s core. Simulating the loss of two of these species in the network structure produced effects similar to the simulated extinction of the two equally structurally important native plant species, emphasizing the utmost importance of these species in maintaining the network structure.

Our investigation into plant-hummingbird network dynamics within an urban garden provides insights that contrast with previous studies examining the impact of exotic plants on ecological interactions. While earlier research, such as Aizen, Morales & Morales (2008) and Maruyama et al. (2016), consistently highlighted the strong attraction of floral visitors to exotic plants and their dominance in interaction networks, our study within the TBG presents distinctive findings. In contrast to the observed dominance of certain exotic plants reported by both referred studies, where they were visited more frequently, our results demonstrate a more balanced interaction between exotic and native plants. While higher-flowering plant species like Grevillea robusta, Melaleuca citrica, and Nicotiana glauca exhibited high connectivity and received significant visits from hummingbirds, native plants within our study site were most preferred by them, as indicated by the obtained preference values (see File S1). This contrasts with the observed ‘usurpation or hijacking’ of interactants reported by previous studies, where exotic plants significantly reduced connectivity among native mutualists. Our findings also diverge from the rapid and pronounced integration of exotic species into interaction networks observed in studies like Montero-Castaño, Vilà & Ortiz-Sánchez (2014) and Seitz, vanEngelsdorp & Leonhardt, 2020. Unlike the immediate changes in network structure due to the introduction of exotic species in other regions, our study suggests a more gradual adaptation of floral visitors to exotic plants within the TBG. This slower integration may indicate a unique resilience or adaptive behavior of native species, thereby maintaining their prominence in the network despite the presence of exotic counterparts. The contrasting patterns observed in our study emphasize the significance of localized ecological contexts in shaping the dynamics of exotic plant integration into ecological networks. While acknowledging the broader trends observed in previous research, our study highlights the resilience and coexistence of native plants within the TBG, providing a nuanced understanding of the interplay between exotic and native species in this specific ecological context.

Exotic plant species, like Jacaranda mimosifolia and Nicotiana glauca, integrate well into the interaction network, providing high floral displays attractive to local hummingbirds. This general pattern of higher flower availability in exotic plants compared to native ones has been previously emphasized by Parra-Tabla et al. (2019) as an inherent requirement for the successful integration of these types of species into native mutualistic networks. These authors suggest that, in addition to this increased attraction, their integration into interaction networks may be facilitated by their great adaptive plasticity to invaded environmental conditions such as climate, soil, and humidity. For instance, recent studies have observed the influence of climatic factors on the presence (Issaly et al., 2023), and on floral attractiveness (Costa et al., 2023) of N. glauca, a pivotal exotic species in the TBG network. In addition, several works have indicated the species’ adaptability to varying pollination environments across native and invaded habitats (Issaly et al., 2020; García et al., 2020; Costa et al., 2023). These findings illuminate the intricate interplay between climatic factors and the presence of key exotic species within mutualistic networks. In the TBG, favorable conditions for the growth and reproduction of these plants are maintained throughout the year through activities such as artificial irrigation, pruning, and/or fertilizer application, making them a consistent resource for floral visitors. In this way, their spatial and temporal constancy contrasts with the ephemeral availability of many native species in the area, which typically have short life cycles and much lower floral displays compared to exotic species. These differences undoubtedly have a direct effect on visitor attraction.

The interaction network demonstrates a nested structure (NODF = 76.59), where specialist plants and hummingbird species interact with well-defined subsets, interacting further with generalists. High nestedness has been previously reported in plant-pollinator networks invaded by exotic plants (NODF values ranging from 21.7 to 56.6; e.g., Albrecht et al., 2014; Díaz Infante, Lara & Arizmendi Mdel, 2020). Exotic plants significantly contribute to network heterogeneity and attract a variety of hummingbird species. In particular, 13 plant species showed a greater contribution to the nesting of the network, with 11 of them being exotic species. This represents more than half of the recorded exotic plants (20 species). These exotic plants significantly contribute to the network’s heterogeneity and asymmetry. Many of the exotic plant species receive a large number of visits from all visiting hummingbird species, maintaining a weak and generalist dependency among the interactors. This pattern observed in our study area, influenced by exotic plants, has also been reported in previous studies. For instance, Valdovinos (2019) analyzed a community of plants and their pollinators in a sub-Andean region of central Chile. These authors demonstrated that invasive (exotic) plants are important for the persistence of the pollination network and for maintaining its nested structure. This structure can even be lost or collapsed after their simulated removal in extinction models. Additionally, Albrecht et al. (2014) conducted a review of 40 plant-pollinator interaction networks from seven different European sites. Their results showed that exotic plants have higher levels of generalization with respect to their pollinators compared to natives. The consequences of this on the network’s topology are that, instead of displacing native species from the network, exotic plants attract pollinators to the invaded modules and tend to play new important topological roles (i.e., network centers, module centers, and connectors). Moreover, they cause role changes in native species, creating larger modules that are more interconnected. These authors also pointed out that the altered interaction structure of invaded networks makes them more robust against simulated random secondary species extinctions, but more vulnerable when exotic plants, typically highly connected, go extinct first. The results from these studies, along with those obtained in our research, provide insight into how dynamic plant-pollinator interaction networks can be. These networks are systems where the arrival of new interactors can not only alter the network structure but also influence the processes of reciprocal selection among the interactors.

Our study’s identification of four distinct modules within the hummingbird-plant network in the TBG aligns with the number of modules (mean ± e.e.: 5.0 ± 0.26) previously reported for other pollination networks invaded by exotic plants (e.g., Albrecht et al., 2014), and prompts an exploration into the factors influencing these modules. Interestingly, although there are more species of exotic plants (20) than native ones (eight), one of the four modules found is dominated by native plants. This indicates that, despite their lower quantity, the prominent role of native plants in the interaction network is not entirely overshadowed by exotic plants. While initial observations may suggest similarities in floral characteristics among modules, a deeper investigation of pollinator attributes and phenological aspects could elucidate their determinants. Modules, as delineated in our research, signify species groups with stronger interactions within themselves than across the network (Carstensen, Sabatino & Morellato, 2016). This insight has allowed us to highlight potential shared pollination traits, referred as pollination syndromes, among plants within these modules, likely attracting similar types of visitors. Our findings echo observations by Santiago-Hernández et al. (2019) in the Chamela-Cuixmala Biosphere Reserve in Mexico, where comprehensive pollination interactions networks revealed distinct characteristics compared to networks based solely on floral visitors. However, while floral traits may appear analogous among modules, variables like pollinator preferences, behaviors or differences in phenology could significantly contribute to these distinct modules. Factors related to pollinator characteristics—such as foraging behavior, preferences for certain floral traits, or temporal visitation patterns—might intricately shape these modules within the network. Therefore, our study prompts a deeper exploration into the potential role of pollinator characteristics and phenology in elucidating the underpinnings of these modules. By investigating these aspects further, we aim to uncover how specific traits or timing-related interactions might contribute to the observed integration of both native and exotic plants within these network modules.

The importance that exotic plants within modules, alongside native plants, can have in the interaction networks with hummingbirds can be enormous. For example, Maruyama et al. (2016) evaluated the role of exotic species in the structure of 21 quantitative plant-hummingbird networks throughout the Americas. One of their most interesting findings showed that larger exotic plant species with more floral display had higher connectivity values between modules compared to smaller exotic plant species. Therefore, they suggest that exotic plants with a greater floral display distribute their interactions more widely among modules in the networks, acting as connectors within these networks. This is significant because connectors imply mechanisms that eliminate boundaries between modules, thereby affecting the dynamics of a network (Albrecht et al., 2014). This pattern is evident in our study, as exotic plant species with the largest floral display in each of the four modules received the highest number of visits from hummingbirds, playing a crucial role in the network structure.

Identifying core and peripheral species in the interaction network emphasized the distinct roles of exotic and native plant species. For example, out of the 28 plant species in the network, seven form a dense core, including four exotic plants. The remaining 21 plant species are peripheral in the interaction network and have limited significance in terms of connectivity both between and within the network’s modules. This observation remarkably aligns with the core-periphery dynamics elucidated by Padrón et al. (2009) and Albrecht et al. (2014), showcasing the pivotal roles of certain species in fostering network connectivity while relegating others to peripheral positions. Simultaneously, it corroborates Aizen, Morales & Morales’s (2008) proposition that invasive species possess attributes—lush flowers and abundant resources—propelling them into network hubs or super generalist roles. The identification of these exotic plants within the core resonates deeply with the theories put forth by Chittka & Schürkens (2001), underlining how exotic species, through increased interaction strength, can hijack network interactions, elevating their centrality and influencing the network’s original dynamics. This integration of exotic species into the core substantiates Bartomeus, Vilà & Santamaría’s (2008) observation that such plants, as super generalists, occupy central roles, impeding other species from assuming similar positions within the ecological community. Consequently, our findings not only substantiate the core-periphery structure in ecological networks but also underscore the significant influence of invasive or exotic plant species, as highlighted by the above-referenced studies, in shaping the structure and functionality of plant-pollinator networks. This amalgamation of empirical evidence underscores the complex interplay between native and exotic species in ecological networks, shedding light on how such dynamics impact the broader ecosystem functioning.

The network approach allows simulations to evaluate community stability and propose measures to minimize disturbance impacts. In our study, simulating the removal of highly connected exotic and native plant species resulted in a minor, approximately 2% loss of interactions. This suggests that the network remains robust even after such removals (Memmott, Waser & Price, 2004). Our findings indicate the robustness of the network’s structure between plants and hummingbirds in the TBG against simulated extinctions of both native and exotic plant species, with an overall high robustness score (R = 87.27). Despite the central roles played by various exotic plant species in the network structure, their simulated removal does not seem to significantly impact interaction loss. This pattern underscores the resilience of the network even when key exotic species are removed. This result aligns with what was reported by Vitorino, da Frota & Maruyama (2021), who compared the importance of native and exotic plant species for the robustness of a plant-hummingbird interaction network in an urban area in Brazil. These authors point out that since exotic plants were only a minor component in the studied network, their removal did not significantly affect the network’s robustness. Contrary to that study, in our work, the removed exotic plants were core species in the network, yielding similar results.

Although with some limitations, as we only sampled a single study area, our study indicates that phenological dynamics of floral resources, both native and exotic plants, reshape hummingbird plant-networks. While our study specifically focuses on an urban garden, we acknowledge the potential value of comparative analyses with natural systems or varied land-use gradients. Such comparisons could indeed enrich our understanding by highlighting differences in network structures, species compositions, and ecological functioning between transformed and natural habitats. Future investigations expanding this study’s scope to encompass a gradient of land-use types could yield valuable insights into the resilience of ecological networks across different environmental contexts.

While recognizing potential positive effects of exotic species, caution is necessary as they can outcompete natives (Bartomeus, Vilà & Santamaría, 2008; Staab, Pereira-Peixoto & Klein, 2020). Prioritizing native plant species is essential for urban green areas, promoting biodiversity and sustaining rich interactions (Aronson et al., 2014; Duguay, Eigenbrod & Fahrig, 2007; Seitz, vanEngelsdorp & Leonhardt, 2020). Urban green areas act as reservoirs of biodiversity, encompassing interactions between plants and visiting hummingbirds.

Conclusions

Our research provides valuable insights into the complex dynamics of plant-hummingbird interactions within an urban green area. Key findings challenge prior assumptions and underscore the significance of native plants in shaping these ecological networks. The presence of asynchronous flowering among native and exotic plants ensures continuous floral resources for hummingbirds throughout the year. Contrary to previous claims of exotic plant dominance, our study reveals a balanced interaction between exotic and native species. Notably, native plants exhibit resilience, maintaining prominence in the ecosystem while exotic species gradually integrate into the network. Exotic plants play crucial roles in the network’s structure, attracting diverse hummingbird species and fostering connectivity between modules. Simulations showcasing the network’s robustness against the removal of exotic key species, known as invasives such as Nicotiana glauca, highlight its resilience to disturbances. However, caution is advised regarding potential competition for pollinators between exotic and native species, emphasizing the conservation importance of prioritizing native plants for urban biodiversity. In summary, our study advances our understanding of plant-hummingbird interactions, emphasizing the pivotal role of native species in sustaining rich ecological networks within urban green areas like the TBG. These findings contribute to a more nuanced perspective on how exotic and native species coexist and shape intricate ecological dynamics.

Supplemental Information

Supplemental Information 1 Detailed data on flower and hummingbird abundance, network core analyses, hummingbird visits and preference index, and the interaction matrix corresponding to the main research.

Supplemental Information 2 Monthly variations in flowers and hummingbirds recorded at the TBG (Tizatlán Botanical Garden, Tlaxcala, México).

The plots illustrate the monthly fluctuations in recorded flower (A and B) and hummingbird (C) abundance observed over a year. Each data point represents the count of flowers and hummingbirds per species recorded during specific months throughout the year. The line in the plot depicts a smoothed trend, providing an overview of the trend in flower and hummingbird abundance over the entire year.

We gratefully acknowledge Citllalli Castillo-Guevara, Mariana Cuautle, Claudia I. Rodríguez-Flores, Valeria Paiaro and one anonymous reviewer for providing useful comments on previous versions of the manuscript. This work constitutes partial fulfillment of M.S.S.’s master degree requirements at UATx.

Additional Information and Declarations

Competing Interests

Author Contributions

Data Availability

The authors declare that they have no competing interests.

Monserrat Sánchez Sánchez conceived and designed the experiments, performed the experiments, analyzed the data, prepared figures and/or tables, authored or reviewed drafts of the article, and approved the final draft.

Carlos Lara conceived and designed the experiments, performed the experiments, analyzed the data, prepared figures and/or tables, authored or reviewed drafts of the article, and approved the final draft.

The following information was supplied regarding data availability:

The raw measurements are available in the Supplemental File.

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
