# Peer review of "Exotic and native plants play equally important roles in supporting and structuring plant-hummingbird networks within urban green spaces"

_PeerJ, doi:10.7717/peerj.16996_

## Round 0.1 · original submission · Major Revisions

I have received two reviews both of which suggest that revisions need to be made to your manuscript. I'd like to draw your attention to comments made by reviewer #1 who asks you to precise more clearly the aims of your study, noting that you may not be able to test a difference between native and alien use because you do not have a controlled study design. Therefore, please be objective about what you can and cannot conclude in this study. There is no problem with conducting a descriptive study using observation data, but you must be clear about your approach and realistic in what you can conclude using your study design. I look forward to receiving your revised manuscript.

Reviewer 1 ·

Basic reporting

The language is sufficient with only a few typos. The literature is sufficient. The figures are mostly good quality, but see specific comments below.
The hypotheses are weak.

Experimental design

The design is adequate. There was good temporal and spatial coverage within the urban garden. However, it is lacking a comparison or control area. There is consistent data collection. The statistics are sufficient for the type of data. Except that month is a random effect. How many data points are there in a month? Is it number of recorded hummingbirds per species (line 206-207)?
The research question is not well defined. This study seems exploratory and there is no clear question. The study describes a plant-pollinator network in a certain location. It is not clear how this contributes anything new to the research field, other than more data. This lack of direction is also seen in the title, which is a bit vague, and abstract. The title would be more informative if it said something about the question being addressed or about the main finding. Furthermore, What is the value of studying a network in an urban garden? It is a highly transformed land-use where one does not expect to find natural interactions. What is the value of studying this alone, without comparison to natural systems or a gradient of land-uses? Please clarify what knowledge gap you are addressing.
The research prediction is that alien species will dominate the structuring of the interaction network. However, 20 out of 28 plants were exotic species, thus obviously the exotics’ contribution dominates. What would be more useful is comparing situations where the abundances of indigenous and alien plants are equal, or changes so that one can see the effect. The authors need to correct and improve the research question of this study.
Further motivation regarding the unsatisfactory research question follows.
Line 100: did you assess it at different abundance levels? The study did not include a control: an area with no exotic species.
Line 149-152: As mentioned earlier, abundance of the exotic plants probably determines their effect. If you want to determine if exotics have a greater effect on network structure, then the abundance of exotics and natives should be equal. Otherwise, all you will be able to say is that the most abundant plants structure the network, irrespective of their origin.

Line 195: does this mean that only the abundance of visited plants were recorded, and not the floral abundance of all ornithophilous plants? Why?

Validity of the findings

The results are described accurately. The result interpretation could improve. Multiple times in the Discussion, there is too much description of other studies and not thorough interpretation of the current study’s findings.
The conclusions are not linked to the original research question. Line 495-496: this statement is not supported by the evidence in this study and are not appropriate conclusions for this study. Please rather make a statement referring to the research question, or what answer was found to the research question posed.

Additional comments

General comments

Tables and figures
Table 1 formatting does not conform to conventions.
Fig 1A and B. Why are the months not on the x-axis? Readers cannot compare the abundance between months in this format. Are these like violin plots on their side? Does the height of the ridge show the number of plants with flower abundance in that abundance category? And what does ridge height mean in Fig 1B? Please explain. Maybe it's just me, but I don’t find these figures intuitively understandable. The ‘overlap’ mentioned in line 272 is not visible from this figure. Based on this figure, exotics have only slightly higher flower abundance than natives, so I find it hard to believe that it is significantly higher. In Fig 1A there is an extra blue box in the key that does not make sense.
Fig 3 legend: it would be useful to add that the circles for hummingbirds are labelled.
Fig 4 legend: give a brief explanation of what negative values mean. Is it decreasing the nestedness?
Figure legends. “at the study site” is not useful information, rather give the city or garden name here.

Points for clarification
Be cautious with the use of ‘Exotic’ and ‘invasive’ in the Introduction, and the interpretation of studies of these different groups. Your study did not include invasive species, or at least did not specify them.
Line 38: It is stated that flowering overlapped. Do you mean flowering of different plant species?
line 40: What is meant with “hummingbird abundance fluctuated”? Do you mean a species’ abundance changed over time, or that the overall hummingbird abundance increased and decreased through the year?
Line 204: which error distribution family was used? Poisson?
Line 209: please specify if the random factors were intercept or slope random effects.
Fig 4: where is the description of contribution to nestedness, of which results are described in Line 300-305? In the methods I find a description of modularity contribution, but I can’t find this in the results.
Line 322: Nicotiana glauca and Melaleuca citrina are exotic species, not native.
Line 344: remove extra comma
Line 417-426: Instead of describing the other study in detail, interpret you own results. Do you see common traits within modules or not?
Line 472-480: some repetition of information

·

Basic reporting

I found the manuscript very well written and easy to read. Introduction is well structured. Literature is well referenced and relevant.
Results are relevant for the hypothesis, well presented with high quality, well labelled, figures accordingly. However, in addition to the overall temporal variation of plants and pollinators showed in Figure 1, it would be interesting to observe how the abundance of each plant and hummingbird species varies over time. Although this information, which is pointed out in the discussion (e.g. lines 332-333, 379-381), is available in the Supplementary Material, I suggest it is graphically showed.

Experimental design

Aims and predictions of the study are clearly exposed. However, I suggest that authors improve the description at lines 141-143 to provide more justification for their study (specifically, they should expand upon the knowledge gap being filled).
Methods are described with sufficient detail and information to be replicated.

Validity of the findings

Discussion is very interesting and complete. However, in several parts, authors make an exhaustive description of other studies’ results (e.g., lines 344-368, 447-452, 455-470). Instead of that, I suggest they focus in the interpretation of their own results, contrasting them with those found in previous works and mentioning only those aspects that allow to discuss their findings. In my opinion, this is the most important point that should be addressed by the authors to improve the manuscript.
In the conclusion, other issue that might be added is that even removing the exotic plant species that are known invasive (such as N. glauca) the network of interactions between plants and hummingbirds would be maintained. This is an important study contribution to management of the green urban area TBG.

Additional comments

This is an interesting study and has considerable potential to contribute to the literature on native and exotic plant-pollinator interactions. Through records of plant-hummingbird interactions over a year-long period in an urban garden at Tlaxcala, Mexico, the study explores the effects of exotic plants on the structure of the interaction networks hummingbirds establish with native plant species. The information provided by this study is important from the conservation perspective, as it can contribute to the proper management of exotic plants in anthropized green spaces.

Particular observations

Lines 322 and 324: Be careful, the order of the words “native” and “exotic” is interchanged! In line 322 it should say “the two core exotic plant species”, while in line 324 it should say “native plant species”.

Lines 373-376: This could be discussed taking into account recent works that found an effect of climatic factors on the presence (Issaly et al. 2023), and on floral attractiveness (Costa et al. 2023), of one core exotic species in the TBG network (i.e., Nicotiana glauca). In addition, several studies have found that this species is able to adapt to changes in pollination environments between native and invaded habitats (Issaly et al. 2020; García et al. 2020; Costa et al. 2023).

Issaly, E. A., Baranzelli, M. C., Rocamundi, N., Ferreiro, A. M., Johnson, L. A., Sérsic, A. N., & Paiaro, V. (2023). Too much water under the bridge: unraveling the worldwide invasion of the tree tobacco through genetic and ecological approaches. Biological Invasions, 1-19.

Costa, A., Moré, M., Sérsic, A. N., Cocucci, A. A., Drewniak, M. E., Izquierdo, J. V., ... & Paiaro, V. (2023). Floral colour variation of Nicotiana glauca in native and non‐native ranges: Testing the role of pollinators' perception and abiotic factors. Plant Biology, 25(3), 403-410.

Issaly, E. A., Sersic, A. N., Pauw, A., Cocucci, A. A., Traveset, A., Benitez-Vieyra, S. M., & Paiaro, V. (2020). Reproductive ecology of the bird-pollinated Nicotiana glauca across native and introduced ranges with contrasting pollination environments. Biological Invasions, 22, 485-498.

García, M., Benítez-Vieyra, S., Sérsic, A. N., Pauw, A., Cocucci, A. A., Traveset, A., ... & Paiaro, V. (2020). Is variation in flower shape and length among native and non-native populations of Nicotiana glauca a product of pollinator-mediated selection?. Evolutionary Ecology, 34, 893-913.

Line 380: Please include a column with the species life form in Table 1 or Supplementary Material.

Lines 423-426: It is not clear which are the factors that, according to authors, determine the network modules found, since it would seem that floral characters are similar between them. Could characteristics of pollinators or phenology explain them?

Lines 472-474: Are these own results? Please clarify.

Lines 483-485: Please mention that unlike what was found in such study, in the present work the exotic plants removed were core species in the network, yielding similar results.

Lines 486-491: Good!

---

## Round 0.2 · Minor Revisions

Thank you for your changes to the manuscript. I have looked at this together with one of the reviews. Although adequate, both the reviewer and I feel that there would be more value added if you could address a few minor issues (detailed below).

With apologies for the delay during this holiday season.

Reviewer 1 ·

Basic reporting

All is sufficient.
Something that would improve the interest level of the study and the prediction formulation is if you could work these questions into the Introduction:
Do alien plants form separate modules in the network?
Do alien plants overlap or complement the flowering time of native plants?

Experimental design

All is sufficient. The research question has been improved.
You could add value to the manuscript by quantifying pollinator preference formally: proportion of visits to plant species/group compared to number of available flowers. With suitable statistics.

Validity of the findings

All is sufficient. The figures and conclusions have been improved.
Perhaps I missed it, but did you compare the nestedness and modularity values to other hummingbird networks? Or other networks, just to put this into context of our current understanding.

Additional comments

Figure 2 legend: please use full name of garden
Fig 3 the light blue modules stands out as being the only native plant dominant one. This was not highlighted in the results or discussion.

---

## Round 0.3 · accepted · Accept

Thank you for comprehensively addressing the comments of reviewer and editor. The corrected manuscript is acceptable for publication in PeerJ. Congratulations!